# Subjective and Objective Cognitive Impairments in Non-Hospitalized Persons 9 Months after SARS-CoV-2 Infection

**DOI:** 10.3390/v15010256

**Published:** 2023-01-16

**Authors:** Inge Kirchberger, Daniela Peilstöcker, Tobias D. Warm, Jakob Linseisen, Alexander Hyhlik-Dürr, Christine Meisinger, Yvonne Goßlau

**Affiliations:** 1Epidemiology, Faculty of Medicine, University of Augsburg, University Hospital of Augsburg, Stenglinstraße 2, 86156 Augsburg, Germany; 2Institute for Medical Information Processing, Biometry and Epidemiology—IBE, LMU Munich, 81377 Munich, Germany; 3Vascular Surgery, Faculty of Medicine, University of Augsburg, 86156 Augsburg, Germany

**Keywords:** outpatients, cognition, memory, depression, COVID-19, long COVID, post COVID

## Abstract

Studies on cognitive problems of persons with mild COVID-19 courses are still lacking. This study aimed to determine the frequency and associated factors of subjective and objective cognitive problems after COVID-19 in non-hospitalized persons. Study participants were examined at the University Hospital of Augsburg from 04/11/2020 to 26/05/2021. The Wechsler Adult Intelligence Scale (WAIS) IV digit span, Stroop Color and Word Test (SCWT), Regensburger verbal fluency test (RWT) and, subjective ratings of memory and concentration were applied. Of the 372 participants (mean age 46.8 ± 15.2 years, 54.3% women, median time after infection 9.1 months), 24.9% reported concentration and 21.9% memory problems. Overall, 55.6% of the participants had at least a mild negative alteration in any cognitive test. The strongest impairments were found regarding memory functions (41.1% mild alterations, 6.2% distinct impairments) and verbal fluency (12.4% mild alterations, 5.4% distinct impairments). SCWT showed negative alterations in no more than 3.0% of the participants. Level of school education, age, and depressiveness emerged as significantly related to the cognitive tests. The number of complaints and depressiveness were significantly associated with subjective memory and concentration problems. It is important to identify mild cognitive impairment in non-hospitalized COVID-19 patients early to offer them effective interventions.

## 1. Introduction

Since December 2019 the novel severe acute respiratory syndrome coronavirus 2 (SARS-CoV-2) that causes coronavirus disease (COVID-19) has reached approximately 514 million cases, with about 25 million of them in Germany [1]. Although the majority of the persons infected with SARS-CoV-2 recover within 3–4 weeks, a number of patients experience persisting symptoms beyond the acute phase of infection [2].

Neurological manifestations have been recognized as part of the SARS-CoV-2 infection [3]. The underlying mechanisms by which SARS-CoV-2 affects the brain may include processes related to inflammation, neuroinvasion, microvascular injury, and hypoxia [4]. Emerging evidence has revealed that the olfactory bulb, thalamus, and brain stem are infected by the virus through a trans-synaptic transfer [5]. The excessive release of inflammatory signals such as cytokines and chemokines, and infection of the astrocytes induce neuroinflammation and neuronal death and may result in further neurodegenerative complications [5].

The most frequent neurological manifestations are headache, myalgias and an impaired sense of smell and taste [6]. In addition, current systematic reviews and meta-analyses suggested an overall high frequency of cognitive impairment after COVID-19 [7,8,9]. However, most of the available studies focus on cognitive impairment shortly after the acute disease or hospital discharge, hospitalized patients, and have only small sample sizes [7,9,10,11,12]. For instance, from 22 studies included in the systematic review by Tavares-Junior et al. [7], only two studies had a follow-up time exceeding 6 months and both had mixed samples of hospitalized patients and out-patients [13,14]. Since only a small proportion of COVID-19 patients have a severe course of disease and require in-hospital treatment (approximately 5% in Germany) [15,16], information regarding the long-term cognitive impairments following a mild COVID-19 infection is needed, but evidence is still lacking [8,17]. Furthermore, it has been proposed that the development of long-term cognitive decline depends on COVID-19 risk factors such as advanced age and comorbidities as well as pulmonary, vascular and neurologic pathological processes and treatment course [18]. However, studies which have analyzed associated factors of cognitive impairment also in persons with mild COVID-19 are scarce [19,20]. Overall, comprehensive knowledge on the long-term cognitive consequences of COVID-19 is of utmost importance, since cognitive impairments have been shown to negatively affect working ability and health-related quality of life [21]. Thus, future studies should promote the identification of individuals susceptible to the development of cognitive impairments after COVID-19 in order to offer appropriate interventions [22].

Therefore, this study aimed to investigate the frequency, severity and associated factors of objective and self-reported cognitive difficulties post COVID-19, using data from a prospective observational study in Augsburg, Germany.

## 2. Material and Methods

### 2.1. Design and Study Population

The present study used data from the Corona Thrombosis Study (COVID-T), a prospective single-center observational study evaluating the consequences of COVID-19 on the vascular system and carried out in Augsburg, Germany. The study sample was recruited from the population living in the city and the county of Augsburg. The public health departments identified eligible persons with past COVID-19 confirmed by positive polymerase chain reaction (PCR) testing and sent out a total of 1600 postal invitations for study participation between 21 October 2020 and 6 November 2020. The potential study participants were invited for clinical examinations and further assessments that were performed at the University Hospital of Augsburg from 4 November 2020 to 26 May 2021. From the 1600 invited persons, a total of 525 (32.8%) participants could be enrolled in the study. Of these, 89 participants with a follow-up (FU) time of less than three months, 56 hospitalized participants, 4 participants with a history of stroke and one participant who had not completed any cognitive test were excluded, leaving 372 participants for the statistical analysis.

The study was approved by the ethics committee of the Ludwig-Maximilians Universität Munich and was performed in accordance with the Declaration of Helsinki. Written informed consent was obtained from all participants.

### 2.2. Survey Data

Data was collected using a self-reporting questionnaire on a tablet personal computer. The survey included information on socio-demographics, disease history, comorbid conditions as well as symptoms during the acute COVID-19 infection and persisting symptoms.

The participants were asked to complete a self-developed list of 42 symptoms, rating them for their occurrence in the acute COVID-19 phase as well as for the 14 days before the FU examination. If the symptom was currently present, a rating of its severity was requested. Furthermore, the participants were asked to compare the current symptom frequency and severity with the symptom burden experienced in the acute COVID-19 phase. The symptom list covered a number of neurological symptoms, including concentration and memory problems, an impaired sense of smell and taste, headache, vertigo, and sleeping problems.

In addition, standardized questionnaires were used to assess depression, post-traumatic stress disorder (PTSD), and health-related quality of life (HRQOL). Depressive symptoms were assessed with the depression module of the Patient Health Questionnaire (PHQ). The PHQ-9 consists of nine items with response options on a scale from 0 to 3 (never to nearly every day), resulting in a score ranging from 0 to 27. A score less than five can be interpreted as the absence of depressiveness. Values between 5 and 10 constitute a mild degree of depressiveness. Values of 10 and higher can be subdivided into moderate (10 to 14), moderately severe (15 to 19), and severe (20 to 27) depressiveness [23].

PTSD was measured using the revised Impact of Event Scale (IES-R). The IES-R consists of three subscales made up of 22 items. Based on a weighted summation of the subscale scores, a total score can be calculated which indicates the likelihood of a suspected diagnosis of PTSD. A score of 0 or below indicates no suspected diagnosis of PTSD, whereas scores above 0 suggest a suspected PTSD diagnosis [24].

To assess HRQOL the Veterans RAND 12-Item Health Survey (VR-12) was used. The VR-12 consists of 12 items assessing physical and mental HRQOL with two subscales. The scores range between 0 and 100 with higher values indicating better HRQOL [25].

### 2.3. Cognitive Assessments

Cognitive evaluation covered memory and executive cognitive functions. The cognitive tests were conducted by trained study nurses. To test the working memory (WM) the subtests “forward, backward and sequential digit span” from the Wechsler Adult Intelligence Scale (WAIS) IV were used [26]. To assess the ability to inhibit cognitive interference, the Stroop Color and Word Test (SCWT) with the subtests “word” (W), “color” (C) and “color-word” (CW) was applied [27]. The semantic verbal fluency test is a subtest of the Regensburger verbal fluency test that aims to assess divergent thinking. The study participants were asked to name as many animals as possible within 2 minutes [28].

### 2.4. Data Analysis

Descriptive statistics were used to present the demographic, psychosocial, and clinical characteristics and the cognitive test results. The frequency and severity of cognitive impairments was determined by comparing patients’ cognitive performance with normative test data derived from different populations available from the respective test manuals. The raw values from the WAIS-IV were transformed into age-corrected standardized value points. A normal distribution with a mean of 10 and standard deviation of 3 was assumed. Cut-off points were 7 for mild or borderline negative alteration of the working memory and 4 for impairment of the working memory. Values higher than 7 indicate no negative alteration [26]. The raw values of the SCWT were transformed into t-standardized age-corrected values, assuming a normal distribution with a mean of 50. The standard deviation of cut-off points were 40 for mild or borderline negative alteration of the inhibition of cognitive interference and 30 for impairment of the inhibition of cognitive interference. Values larger than 40 can be interpreted as no negative alteration [29]. Raw values of the Regensburger verbal fluency test were transformed and age-corrected into percentage ranks with 16% as the cut-off point for a mild or borderline negative alteration of divergent thinking and 2% for an impairment of divergent thinking. Values equal or higher than 17 indicate no negative alteration [28].

Five multiple linear regression models were fitted to determine the association between working memory, cognitive interference, semantic verbal fluency and a number of possible demographic, psychosocial and clinical covariables. In addition, the association between the subjective ratings of present problems with concentration or memory and demographic, psychosocial and clinical covariables was determined by two multiple logistic regression models. The covariables included in the models were selected based on current literature [7,8,9].

The assumptions for multiple linear regression analyses were tested using scatterplots and Q-Q plots to confirm the linearity of associations, normal distribution of the residuals and homoscedasticity. Cook’s distances and leverage diagnostic plots were used to check for leveraging outliers. Variance inflation factors were used to identify multicollinearity and Durbin–Watson tests were calculated to determine autocorrelation.

For statistical tests, an alpha level of 0.05 was defined. The statistical analyses were performed using SAS-Studio.

## 3. Results

### 3.1. Demographic Characteristics

The demographic and clinical characteristics of the participants are shown in Table 1. The mean age of the enrolled participants was 46.8 ± 15.2 years with 170 (45.7%) men and 202 (54.3%) women. The median time between SARSCoV-2 infection and FU examination was 9.1 months.

Among the participants, 156 (43.0%) had at least one comorbidity. The most common comorbidity was hypertension (*n* = 71, 19.1%), followed by depression (*n* = 32, 8.6%) and autoimmune disease (*n* = 30, 8.1%). The most common symptom was tiredness or exhaustion with a prevalence of 83.8% in the acute phase and 34.0% at FU. Headaches, sleeping problems and disturbance of taste or smell were reported by more than 20% of the participants at the FU examination. At the FU examination, mild depressiveness was found in 103 participants (27.7%), whereas 48 (12.9%) had moderate to severe depressiveness. Furthermore, 15 participants (4.1%) showed a suspected diagnosis of PTSD.

### 3.2. Cognitive Impairments

Results of the evaluation of subjective and objective cognitive problems are shown in Table 2. A total of 164 participants (44.2%) stated having had difficulties concentrating during the acute COVID-19 infection, and 92 participants (24.9%) reported having had difficulties with concentration in the two weeks prior to the FU examination, with the majority having either slightly (*n* = 38, 41.8%) or moderately (*n* = 20, 22.0%) severe problems. Memory problems during the acute COVID-19 infection were reported by 90 participants (24.3%). A total of 81 participants (21.9%) reported having had memory problems in the two weeks prior to the FU examination, with the majority having either slightly (*n* = 31, 38.3%) or moderately (*n* = 21, 25.9%) severe memory problems. Compared with at the time of the acute COVID-19 infection, most of the participants rated their current concentration problems as being equally severe/frequent (*n* = 20 (25.6%) or even more frequent/severe (*n* = 28, 35.9%). Similarly, memory problems were considered as being persistent by 10 participants (19.6%) or more frequent/severe by 20 participants (39.3%).

Among the participants, 23 (6.2%) obtained a score signaling impairment in the WAIS-V test, and 153 (41.1%) showed a mild/borderline negative alteration in terms of memory function. In 23 participants (6.2%), an impairment was observed, and in 47 (12.6%) a mild/borderline negative alteration of the verbal fluency was observed on the RWT test. Fewer participants showed impairment or negative alterations in the SCWT tests, with one participant (0.3%) who had an impairment, and up to 10 participants (2.7%) with mild/borderline negative alterations.

### 3.3. Variables Associated with Cognitive Impairments

The multivariable linear regression models for the cognitive tests are shown in Figure 1 and Appendix A. All five models included the same independent variables: sex, age, education, FU time, sum of complaints, concentration problems at FU, memory problems at FU, depression, PTSD, mental HRQOL, and neurological symptoms at FU (disturbance of smell or taste, headaches, vertigo, sleep problems).

These variables explained 8% (adjusted R^2^) of the WAIS-IV variance. Participants with equal or less than 9 years of school education had significantly lower WAIS-IV scores (indicating more memory problems) compared with participants with more than 9 years school education (β = −1.38, *p* < 0.0001). In addition, women were more likely to have memory problems than men (ß = −0.62, *p* = 0.0125).

The regression model for verbal fluency showed 13% explained variance of the RWT scores. Besides years of school education, a higher age (β = 0.71, *p* < 0.0001), and higher depression scores (PHQ-9) (ß = 1.51, *p* = 0.0332), worse mental HRQOL (VR-12) (ß = 0.57, *p* = 0.0266) and lower PTSD scores (IES-R) (ß = −3.76, *p* = 0.0297) were associated with lower RWT scores, indicating impaired verbal fluency.

The linear regression model for SCWT–W has an adjusted R^2^ of 14%. Lower age (β = 0.16, *p* < 0.0001), school education equal or less than 9 years (β = −3.10, *p* = 0.0016) and worse mental HRQOL scores (β = 0.14; *p* = 0.0113) were significantly associated with lower SCWT–W scores. The model for SCWT–C has an adjusted R^2^ of 2%. Similar to the SCWT–W results, participants with school education equal or less than 9 years had significantly lower SCWT–C scores than participants with longer school education (β = −2.81, *p* = 0.00311). In addition, higher depression scores (ß = 0.48, *p* = 0.0191) were significantly associated with lower SCWT–C scores. The regression model explaining the variance of SCWT-CW has an adj. R^2^ of 4%. The variables age (β = 0.05, *p* = 0.0398), school education equal or less than 9 years versus more than 9 years (β = −2.14, *p* = 0.0370), higher depression score (ß = 0.32, *p* = 0.0442) and lower PTSD scores (ß = 2.35, *p* = 0.0441) were significantly related with lower SCWT–CW scores, whereas women (β = 1.97, *p* = 0.0108) had significantly higher SCWT–CW scores than men.

Results of the multivariable logistic regression models for self-reported difficulties concentrating and memory problems at FU are shown in Figure 2 and in Appendix A. Participants reporting concentration difficulties at FU were significantly younger (OR = 0.97, *p* = 0.0142), had significantly more complaints (OR = 1.59, *p* < 0.0001), higher depression scores (OR = 1.15, *p* = 0.0452) and were less likely to report smell disturbance (OR = 0.23, *p* = 0.0186) than participants without concentration difficulties. Moreover, participants who reported concentration problems during acute COVID-19, had an almost 12-fold odds of having concentration problems at FU (OR = 11.81, *p* < 0.0001).

Similarly, the presence of memory problems in the COVID-19 acute phase was the main predictor of memory problems at FU (OR = 7.09, *p* < 0.0001), and more complaints (OR = 1.32, *p* < 0.0001) as well as higher depression scores (OR = 1.25, *p* = 0.0010) were significantly associated with self-reported memory problems.

### 3.4. Sensitivity Analyses

For the cognitive test results and subjective cognitive problems, regression models were modified by including the neurological symptoms reported in the acute COVID-19 phase instead of 14 days before FU examination. The results, however, were very similar (see Appendix A).

## 4. Discussion

The present study examined cognitive functions based on objective tests and self-reported problems in 372 participants at an average of 9 months following a mild COVID-19 infection. Overall, 24.9% and 21.9% of the participants reported having concentration problems and memory problems, respectively. Cognitive test results revealed that 55.6% of the participants had at least a mild negative alteration in any tested cognitive function. The strongest impairments were found regarding memory functions, where 41.1% of the study participants showed mild/borderline negative alterations compared with population norms, and distinct impairments were found in 6.2% of the participants. In terms of word fluency, 12.4% of the study participants had mild/borderline negative alterations and 5.4% had distinct impairments. In contrast, cognitive interference, as tested by the SCWT, was negatively affected in no more than 3.0% (depending on the subtest) of the study participants. Level of school education, age and depressiveness emerged as the most important predictors of cognitive test results. Self-reported memory and concentration problems at follow-up were mainly explained by the presence of these problems in the acute phase. In addition, the number of reported complaints at follow-up and the level of depressiveness were significantly associated with the report of memory and concentration problems.

Overall, it is difficult to compare the results from the present study with previous results, since available studies considerably differ regarding the assessment of cognitive functions, the cut-offs for defining impaired functions, the time between acute COVID-19 and examination, and the clinical severity of COVID-19.

In terms of self-reported cognitive difficulties, the findings from the present study are in line with the results of studies reporting memory problems in 19.5–34.0% and attention difficulties in 24.4–28.0% of the participants 4–15 weeks post COVID-19 [30,31]. A smaller number of cognitive problems was found in a study from Mattioli et al. [32], with attention difficulties reported by 11.6% and memory difficulties by 6.6% of 120 persons 4 months after mild to moderate COVID-19. The present study suggests that self-reported cognitive problems tend to persist and even deteriorate over time. This finding may be partly explained by the ongoing restraining measures due to COVID-19, including social distancing, social isolation and restriction of movements, which the entire population was exposed to [33]. In contrast, Del Brutto et al. [34] found an improvement in cognitive impairments over 1.5 years in patients with mild COVID-19.

In terms of objective cognitive tests, with the exception of Mattioli et al. [32], most prior studies reported cognitive impairments post COVID-19 [21,35]. Jaywant et al. [36] found mild to severe cognitive impairment in 81% of 57 COVID-19 patients 7 days after diagnosis. Similar to the present study, working memory was the cognitive function most strongly affected. Mazza et al. [37] found that 78% of 226 patients showed poor performance in at least one cognitive domain 3 months after hospital discharge; 24% had poor performance in working memory and 32% in verbal fluency. Albu et al. [35] and Miskowiak et al. [21] both reported a cognitive impairment in 59–65% of their participants 2–4 months after hospital discharge. However, the validity of these studies is limited by the small sample sizes, *n* = 30 and *n* = 29, respectively. Lower prevalences of cognitive deficits were reported by Rass et al. [38] (23% of 135 participants 3 months post COVID-19), Negrini et al. [39] (33.3% of 9 participants 44 days after hospital admission) and Almeria et al. [40] (5.7–11.4% (depending on the specific test) of 35 participants 26 days after hospital discharge).

Although a recent systematic review indicated that cognitive impairment can be also found in mild or asymptomatic cases of COVID-19, studies which focus on non-hospitalized persons are rare [8]. Hampshire et al. [18] found significant cognitive deficits in 326 non-hospitalized COVID-19 cases compared with controls. The only study which compared cognitive function of asymptomatic COVID-19 subjects (*n* = 93) with healthy controls found significantly more cognitive dysfunction in the domains of fluency, visuoperception and naming in the COVID-19 group [41].

Of interest is that the regression analysis of the cognitive tests revealed no variables which were consistently and independently associated with cognitive impairments, except poor education level. Since poor education may adversely affect cognition via low cognitive reserves, future studies should be conducted on this specific patient group [5]. In contrast to the results from Almeria et al. [40], the presence of neurological symptoms was not consistently associated with cognitive functions in the present study. Only disturbance of smell function was found to be related with the persons’ subjective rating of concentration problems and worse scores on the SCWT-CW test. However, Almeria et al. [40] also included hospitalized persons, and the cognitive tests were performed shortly after hospital discharge.

In contrast, depression was significantly associated with impairments of executive cognitive functions as well as with subjective memory and concentration problems. It is well known that people with depressive disorders suffer from cognitive impairments; however, the association between depression and cognitive problems is so far rarely investigated in studies on COVID-19 [20,40]. Of interest is that, similar to other studies [35,40,42], subjective cognitive problems were not significantly associated with test results in the present study. Subjective cognitive problems were, however, strongly related with the number of post-COVID-19 complaints and depressiveness, indicating relevant risk groups for post-COVID-19 cognitive impairment. In addition, the failure to objectify subjective cognitive impairments by means of standard cognitive tests underscores the need to define a standard test battery applicable for COVID-19 survivors.

Furthermore, the present study showed that the independent variables included in the regression models explained only a low percentage of the cognitive test variance. Consequently, other factors are suggested to be associated with cognitive impairment after COVID-19. These could include pre-COVID-19 cognitive problems or consequences of the COVID-19 inflammation process, such as pulmonary hypoxaemia and neuroinflammation [18]. Thus, further studies are needed in order to specify the factors contributing to the development of cognitive dysfunction after COVID-19.

## 5. Strengths and Limitations

To the best of our knowledge, the present study is the first which investigated both subjective and objective cognitive functions in a large sample of non-hospitalized persons with COVID-19. Further strengths of this study are the median follow-up time of nine months and the inclusion of participants with confirmed COVID-19 in a defined study region.

However, this study has several limitations. Firstly, the study was lacking a healthy control group. Comparison of test results was based on the published norming populations. Secondly, the participants reported their symptoms during the acute COVID-19 infection phase retrospectively. These statements are, therefore, subject to a possible recall bias. In addition, information on previous cognitive impairments was missing. Furthermore, a selection bias cannot be excluded, since persons with a large disease burden may be more likely to participate in a study than persons without any health impairments after COVID-19. Therefore, the cognitive impairments found in the present study may be overestimated. The variance explained by the independent variables in the regression models of cognitive tests was small, indicating that relevant predicting factors may be missing. Finally, the study participants were infected with the early SARS-CoV-2 variants of concern, and the results may not be transferrable to later virus variants.

## 6. Conclusions

In conclusion, the present study consisting of participants with mild COVID-19 course highlights the need to consider screening for post COVID-19 cognitive impairment also in this large group of affected people. Whether cognitive problems after COVID-19 considerably impair the daily life activities and other domains of quality of life of the affected persons should be investigated in future studies. Such results may support the estimation of treatment needs. Furthermore, since no clear predictors of cognitive impairments could be identified in the present study, the role of neuroinflammation should be further investigated in order to find suitable preventive strategies and specific therapeutic approaches. Meanwhile, people who suffer from post-COVID-19 cognitive problems can be supported by established cognitive trainings. Given the prevalence of COVID-19, digital interventions, which can be widely disseminated, might be a useful approach for those with mild cognitive impairments [16]. For more severely affected individuals, extended cognitive diagnostics and treatment may be provided within in- or out-patient rehabilitation or in specialized post-COVID outpatient clinics.

## Figures and Tables

**Figure 1 viruses-15-00256-f001:**
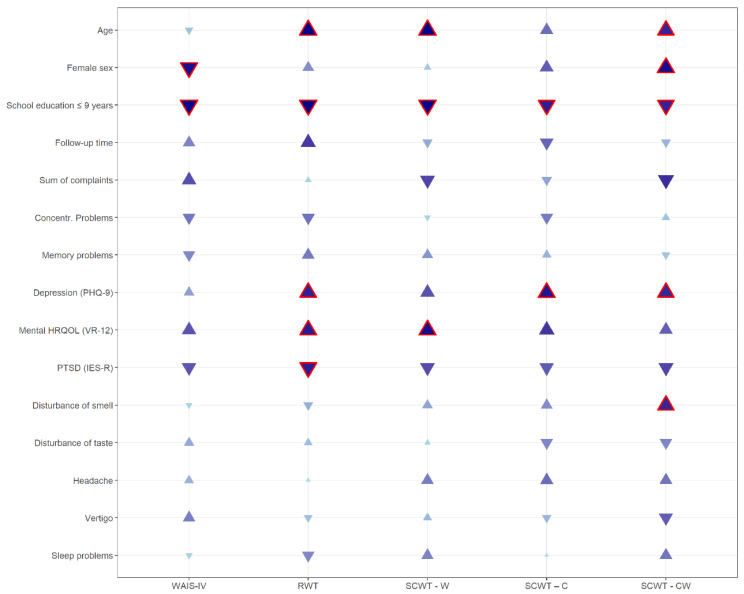
Multivariable linear regression models for the cognitive tests Wechsler Adult Intelligence Scale (WAIS-IV); Regensburger Wortflüssigkeitstest (verbal fluency) (RWT); and Stroop Color and Word Test (SCWT), word (W), color (C), color words (CW). Directions of triangles correspond to the directions of estimates. Triangles outlined in red represent significant *p*-values. The color and size of triangles are based on the *p*-values (i.e., large dark blue triangles represent low *p*-values and vice versa).

**Figure 2 viruses-15-00256-f002:**
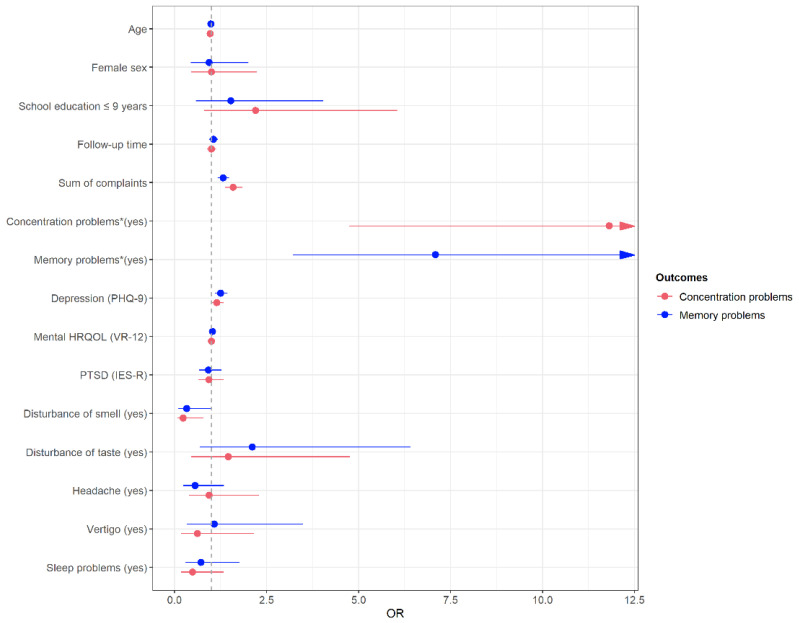
Multivariable logistic regression models for the dependent variables concentration problems and memory problems at follow-up. Odds Ratios (OR) and 95% confidence intervals.

**Table 1 viruses-15-00256-t001:** Demographic and clinical characteristics of the study sample (*n* = 372).

Demographic Characteristics	
**Age** (years) (mean (SD), MIN–MAX)	46.8 (15.2), 18–87
**Sex**	
Female	202 (54.3%)
Male	170 (45.7%)
**School education**	
≤ 9 years	63 (17.0%)
> 9 years	308 (83.0%)
**Marital status**	
Married	246 (66.5%)
Single	98 (26.5%)
Divorced	20 (5.4%)
Widowed	6 (1.6%)
**Smoking**	
Never-smoker	203 (54.6%)
Current smoker	29 (7.8%)
Former smoker	140 (37.6%)
**Clinical characteristics**	
**Comorbidities** (yes)	
Hypertension	71 (19.1%)
Coronary heart disease or angina pectoris	17 (4.6%)
Heart attack	9 (2.4%)
Diabetes	14 (3.8%)
Cancer	17 (4.6%)
Depression	32 (8.6%)
Anxiety Disease	22 (6.0%)
Chronic Bronchitis	21 (5.7%)
Autoimmune Disease	30 (8.1%)
**Sum comorbidities**	
None (0)	213 (57.0%)
At least one (≥1)	156 (43.0%)
**FU time** (months) (median (Q1-Q3))	9.1 (6.0–11.3)
≥ 3–≤ 6	95 (25.5%)
> 6–≤ 9	88 (23.7%)
> 9–≤ 12	111 (32.8%)
> 12–≤ 15	67 (18.1%)
**Neurological symptoms**	
Headache	
Acute COVID-19 phase	244 (65.8%)
FU examination	80 (21.6%)
Disturbance of sense of taste	
Acute COVID-19 phase	233 (62.8%)
FU examination	60 (16.3%)
Disturbance of smell	
Acute COVID-19 phase	228 (61.5%)
FU examination	75 (20.3%)
Vertigo	
Acute COVID-19 phase	130 (35.1%)
FU examination	34 (9.2%)
Sleeping problems	
Acute COVID-19 phase	108 (29.2%)
FU examination	87 (23.5%)
Coordination of movements	
Acute COVID-19 phase	53 (14.3%)
FU examination	16 (4.4%)
Tiredness or exhaustion	
Acute COVID-19 phase	310 (83.8%)
FU examination	126 (34.0%)
Depressive mood	
Acute COVID-19 phase	97 (26.2%)
FU examination	61 (16.5%)
Anxiety or panic	
Acute COVID-19 phase	80 (21.6%)
FU examination	28 (7.6%)
**Sum of complaints**	
Acute COVID-19 phase (median, (Q1–Q3))	13 (9–19)
FU examination (median, (Q1–Q3))	3 (1–7)
**Depressiveness** (PHQ-9) (median (Q1–Q3))	4 (2–7)
<5:	221 (59.4%)
≥ 5–<10: mild depressiveness	103 (27.7%)
≥ 10–<15: moderate depressiveness	38 (10.2%)
≥ 15–<20: moderately severe depressiveness	8 (2.2%)
≥ 20: severe depression	2 (0.5%)
**Post-traumatic stress disorder** (IES-R total score)-	
0: no suspected diagnosis	353 (95.9%)
>0: suspected diagnosis	15 (4.1%)
**Physical HRQOL** (VR-12) (median (Q1–Q3))	52.7 (45.6–55.8)
**Mental HRQOL** (VR-12) (median (Q1–Q3))	50.9 (43.8–56.3)

FU: Follow-up; PHQ-9: Patient Health Questionnaire; IES-R: Impact of Event Scale revised; HRQOL: Health-related quality of life; VR-12: Veterans RAND 12-Item Health Survey.

**Table 2 viruses-15-00256-t002:** Cognitive evaluation: results of subjective ratings of concentration and memory problems, and standard cognitive tests.

Subjective Ratings	
**Difficulties Concentrating**	
Acute COVID-19 phase (yes)	164 (44.2%)
FU examination (yes)	92 (24.9%)
Severity at FU (*n* = 91)	
No impairment	3 (3.3%)
Slightly impaired	38 (41.8%)
Moderately impaired	20 (22.0%)
Fairly impaired	17 (18.7%)
Severely impaired	13 (14.3%)
Frequency/severity at FU compared with acute COVID-19 phase (*n* = 78)	
Much less often/weaker (−3)	9 (11.5%)
−2	11 (14.1%)
−1	10 (12.8%)
Equal frequency/severity (0)	20 (25.6%)
+1	6 (7.7%)
+2	12 (15.4%)
Much more often/stronger (+3)	10 (12.8%)
**Memory problems**	
Acute COVID-19 phase (yes)	90 (24.3%)
FU examination (yes)	81 (21.9%)
Severity at FU (*n* = 81)	
No impairment	6 (7.4%)
Slightly impaired	31 (38.3%)
Moderately impaired	21 (25.9%)
Fairly impaired	11 (13.6%)
Severely impaired	12 (14.8%)
Frequency/severity at FU compared to acute COVID-19 phase (*n* = 51)	
Much less often/weaker (−3)	7 (13.7%)
−2	4 (7.8%)
−1	10 (19.6%)
Equal frequency/severity (0)	10 (19.6%)
+1	6 (11.8%)
+2	9 (17.7%)
Much more often/stronger (+3)	5 (9.8%)
**Cognitive tests**	
**WAIS-IV:** median, (Q1–Q3)	8 (6–9)
≥8: normal	196 (52.7%)
>4–≤7: mild/borderline negative alteration	153 (41.1%)
≤4: impairment	23 (6.2%)
**RWT:** median, (Q1–Q3)	63 (28–87)
≥17: normal	302 (81.2%)
>2–≤16: mild/borderline negative alteration	47 (12.6%)
≤2: impairment	23 (6.2%)
**SCWT–W:** median, (Q1–Q3)	54 (50–60)
≥41: normal	358 (97.0%)
>30–≤ 40: mild/borderline negative alteration	10 (2.7%)
≤30: impairment	1 (0.3%)
**SCWT–C:** median (Q1–Q3)	55 (50–62)
≥41: normal	361 (97.1%)
>30–≤ 40: mild/borderline negative alteration	6 (1.6%)
≤30: impairment	1 (0.3%)
**SCWT–CW:** median (Q1–Q3)	56 (53–62)
≥41: normal	363 (98.6%)
>30–≤ 40: mild/borderline negative alteration	4 (1.1%)
≤30: impairment	1 (0.3%)

FU: Follow-up; WAIS-IV: Wechsler Adult Intelligence Scale; RWT: Regensburger Wortflüssigkeitstest (verbal fluency); SCWT: Stroop Color and Word Test, word (W), color (C), color words (CW).

## Data Availability

The datasets generated during and/or analyzed during the current study are not publicly available due to data protection requirements but are available in an anonymized form from the corresponding author on reasonable request.

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
