# Peer review of "Subjective and Objective Cognitive Impairments in Non-Hospitalized Persons 9 Months after SARS-CoV-2 Infection"

_viruses, 2023, doi:10.3390/v15010256_

Round 1

Reviewer 1 Report

Please find the attachment file for comments details. 

Reviewer 2 Report

Thank you for the opportunity to review the manuscript “Post-COVID-19 subjective and objective cognitive impairments in persons after outpatient treatment” (viruses-2116024).

The authors investigate the frequency, severity and associated factors of objective and self-reported cognitive difficulties post COVID-19, using data from a prospective observational study.

The objectives and the rationale of the study are clearly stated.

However, the authors should work out more clearly why patients with mild courses sought outpatient care (preferably in the introduction).

The interpretation of results and study conclusions are supported by the data.

The Authors also clearly emphasized the strengths of their study and clearly stated the limitations.

Many assessment parameters are used. For the reader would therefore be advantageous: if the central results were summarized in a figure (in results section).

In the conclusion, implications that can be derived from the study should be presented in more detail.

Round 2

Reviewer 1 Report

NO